# Stem Cell Surgery and Growth Factors in Retinitis Pigmentosa Patients: Pilot Study after Literature Review

**DOI:** 10.3390/biomedicines7040094

**Published:** 2019-11-30

**Authors:** Paolo Giuseppe Limoli, Enzo Maria Vingolo, Celeste Limoli, Marcella Nebbioso

**Affiliations:** 1Low Vision Research Centre of Milan, p.zza Sempione 3, 20145 Milan, Italy; paololimoli@libero.it (P.G.L.); celeste.limoli@libero.it (C.L.); 2Department of Sense Organs, Faculty of Medicine and Odontology, Sapienza University of Rome, p.le A. Moro 5, 00185 Rome, Italy; enzomaria.vingolo@uniroma1.it

**Keywords:** autograft, embryonic stem cells (ESCs), growth factor (GF), hereditary retinal disease, induced pluripotent stem cells (iPSCs), limoli retinal restoration technique (LRRT), mesenchymal stem cell (MSC), retinitis pigmentosa, spectral domain-optical coherence tomography (SD-OCT)

## Abstract

To evaluate whether grafting of autologous mesenchymal cells, adipose-derived stem cells, and platelet-rich plasma into the supracoroideal space by surgical treatment with the Limoli retinal restoration technique (LRRT) can exert a beneficial effect in retinitis pigmentosa (RP) patients. Twenty-one eyes underwent surgery and were divided based on retinal foveal thickness (FT) ≤ 190 or > 190 µm into group A-FT and group B-FT, respectively. The specific LRRT triad was grafted in a deep scleral pocket above the choroid of each eye. At 6-month follow-up, group B showed a non-significant improvement in residual close-up visus and sensitivity at microperimetry compared to group A. After an in-depth review of molecular biology studies concerning degenerative phenomena underlying the etiopathogenesis of retinitis pigmentosa (RP), it was concluded that further research is needed on tapeto-retinal degenerations, both from a clinical and molecular point of view, to obtain better functional results. In particular, it is necessary to increase the number of patients, extend observation timeframes, and treat subjects in the presence of still trophic retinal tissue to allow adequate biochemical and functional catering.

## 1. Introduction

Retinitis pigmentosa (RP) comprises a heterogeneous group of hereditary retinal diseases characterized by progressive degeneration of photoreceptors. It primarily and severely affects the rods, with subsequent involvement of cone functions [1,2,3].

Although the etiology is quite variable, the ultimate pathway is progressive photoreceptor cell death by apoptosis, with subsequent retinal atrophy. The prevalence of RP is approximately 1:4000, affecting more than 1 million people worldwide [4].

In X-linked patients, who account for approximately 5–15% of all cases, the phenotype of the disease generally tends to be the most severe. Conversely, patients with autosomal recessive RP, comprising 50–60% of cases, and patients with autosomal dominant RP, which is responsible for 30–40% of cases, show a better visual prognosis, slower progression of the disease, and longer maintenance of central vision. A large number of mutations in more than 80 different genes are known to be the major cause of RP [1,2,3,4].

The etiopathogenesis of RP cannot be explained by genetics alone, because there are other mechanisms that cover various biological aspects: Trophism, oxidation, inflammation, immune response, vascularization, and apoptosis [5].

In the majority of cases, visual impairment usually begins with night blindness and progresses to the restriction of peripheral vision. Macular degeneration usually occurs only at the very end stage of the disease, and may also culminate in the loss of central vision [1,2,6].

The suspect of the disease, caused by visual concerns, can be confirmed by specific examinations, such as visual field testing, full-field electroretinogram (ERG), and optical coherence tomography (OCT) [7,8].

To date, the disease has no curative treatment, but new therapeutic options are being actively developed, involving implanted retinal prosthetic devices, gene therapy, and cell therapy, to replace or restore defective cells [9,10,11,12]. Cell preservation is being actively investigated, especially as regards the neurotrophic, antiapoptotic, hemorheologic, and immunomodulatory actions of growth factors (GFs) and cytokines, which can be used directly or in a cell-mediated way, targeting the residual retinal cells [9,10,11,12,13,14,15].

The therapeutic aim is to slow down or prevent the death of photoreceptors by delivering embryonic stem cells (ESCs), induced pluripotent stem cells (iPSCs), and mesenchymal stem cells (MSCs) to precise target locations in the eye [16,17,18,19,20,21]. 

ESCs, iPSCs, and MSCs are capable of self-renewal and display multipotency, i.e., the ability to differentiate into all cells derived from any of the three germ layers.

MSCs can be obtained from different sources: Umbilical cord, peripheral blood, bone marrow, and adipose tissue [22,23,24]. They therefore play a key role in organogenesis and remodeling, as well as in tissue repair and reactivation synaptic connections by means of GFs, and can therefore enhance the formation of new functional conditions [24,25]. Other positive aspects are the immunosuppressant function and the inhibition of proinflammatory cytokine release [26,27,28].

As demonstrated by clinical and preclinical studies, MSC administration does not require immunosuppression, nor does it induce neoplastic transformation; moreover, it is associated with a significant restoration of the visual system through cell-mediated therapeutic mechanisms [21,28,29,30].

Recently, the Limoli retinal restoration technique (LRRT) has been developed as a potential therapy for currently untreatable retinal disorders. This surgical technique is a variant of Pelaez’s intervention, wherein only orbital autologous fat is transplanted in the subscleral space [31,32,33]. The technique exploits the use of GFs to create an environment conducive to the neuroenhancement of still functioning retina [34,35]. The source of autologous GFs in LRRT is an implant of certain cell types of mesenchymal origin, such as adipose stromal cells, adipose tissue-derived stem cells (ADSCs) contained in the stromal vascular fraction of adipose tissue, and platelets (PLTs) obtained from PLT-rich plasma (PRP) prepared from fresh whole blood by centrifugation (Figure 1) [31,32,33,34,35].

The photoreceptors also receive mediated trophic action from potentially improved conditions of Müller cells, retinal pigment epithelium (RPE) cells, and retinal microcirculation.

In order to evaluate the prognosis of treated RP patients, we hypothesized that the larger the residual cell number is, the greater the interaction between the autograft and the membrane receptors of chorioretinal cells, cellular activity, and, ultimately, the improvement of visual performance.

The primary aim of this prospective, pilot clinical study was to evaluate whether autologous stem cell transplantation in patients with RP, via LRRT surgery, may be beneficial to retinal restoration. Furthermore, the secondary aim was to evaluate prognostic factors to identify the time and tests needed to allow appropriate surgical intervention in those affected with RP.

## 2. Materials and Methods

Approval by the Institutional Review Board of the Low Vision Academy (No: 2016/A101, date: 1 October 2016) was obtained, and the study was conducted in accordance with the tenets of the Declaration of Helsinki. All of the patients were individually instructed on the methodology of the study, and written informed consent was obtained from all participants included.

Six patients signed informed consent again to carry out the same intervention in the contralateral eye. In this study, 15 patients with RP were included if they had:Clinical diagnosis of RP based on a history of night blindness, visual field constriction, abnormalities on ERG testing, and specific ophthalmoscopic findings;Age ranging from 19 to 86 years;Normal intraocular pressure;Visual acuity for near (close-up) vision between 7 and 64 points (pts) in order to avoid difficult evaluations for both low visus (>64 pts) and normal visus (6 pts);Transparent lens;Signature of the informed consent;Retinitis pigmentosa pattern that can be detected at the macula.

The exclusion criteria were the following:Hypermetropy or myopia with spherical equivalent ≥6 diopters;Existence of keratoconus, cataract, cystoid macular edema, keratitis, uveitis, etc.;Other ocular diseases, for example, glaucoma, optic neuritis, ocular trauma, etc.;Lack of patient compliance due to medical conditions, such as Parkinson’s disease, diabetes mellitus, hypertension, vasculitis, hypovitaminosis, multiple sclerosis, epilepsy, or other systemic acute or chronic diseases.

A complete ophthalmologic examination was performed, including the measurement of visual acuity for far and near distance: Best corrected visual acuity (BCVA) measured by early treatment diabetic retinopathy study (ETDRS) charts at 4 m in logarithm of the minimum angle of resolution (logMAR) units and close-up visus (pts); slit-lamp biomicroscopy with and without dilatation; applanation tonometry; and fundus examination.

All eyes recruited for the study cohort were divided into two groups. The division was based on foveal thickness (FT) measured with spectral domain OCT (SD-OCT). For this purpose, a cut-off of ≤190 μm was used. Frequently, in RP patients, the retinal cell population is small, foveal structures are often dystrophic, and the photoreceptor/retinal pigment epithelium/Bruch’s membrane/choriocapillaris complex is no longer recognizable. In those patients with thicker FT, the retinal cell population is large, foveal structures are still intact, and the photoreceptor/retinal pigment epithelium/Bruch’s membrane/choriocapillaris complex is recognizable. Consequently, the subjects with FT ≤190 μm were included in group A-FT, whereas subjects with FT >190 μm were included in group B-FT. At baseline (T0) and 6 months after surgery (T180), the ophthalmologic evaluation and the following exams were performed on each patient: SD-OCT, using the Cirrus 5000 (Carl Zeiss Meditec AG, Jena, Germany); microperimetry (MY) by means of Maia 100809 (CenterVue S.p.A., Padua, Italy); ERG test using Retimax (C.S.O. S.r.l., Scandicci, Italy), an ocular electromedical system, in accordance with the 2009 guidelines of the International Society for Clinical Electrophysiology of Vision (ISCEV) [7]. Comprehensive ophthalmic examination and LRRT surgery [31,32,33] were carried out on all patients by a single retinal specialist (PGL) according to our technique, as detailed in the literature and presented in a video in 2018 [31,33]. Briefly, 10 mL of fat tissue was harvested manually from the patient’s abdominal subcutaneous layer with a cannula connected to a syringe. Pure stromal vascular fraction (SVF) of fat tissue, very rich in ADSCs, was separated from blood, fat, oil, and liquid by centrifugation. Peripheral blood was collected in a Regen-BCT tube (RegenKit; RegenLab, Le Mont-sur-Lausanne, Switzerland) for PRP gel preparation and was centrifuged for 5 min.

A suprachoroidal pocket was created in the patient’s eye to place the cell graft and was filled with a precise amount of ADSCs and SVF. A 5-mm-deep scleral door was opened by radial hinge in the infero-temporal quadrant at 8 mm from the limbus. Orbital fat was collected from the space above the inferior oblique muscle. The adipose flap obtained was placed gently on the bed and sutured with choroidal vicryl 6/0 at the proximal edge of the door. Subsequently, 1 cc of PRP was injected in the stroma of the peduncle using a 25-gauge cannula. As a result, an autograft composed of fat cells, ADSCs from SVF, and PRP was carried out.

### Statistical Analysis

Data are presented as mean ± standard deviation (SD); minimum and maximum (min–max) values are reported as well. Mixed regression models with robust errors were applied to analyze the difference between the two groups according to the foveal thickness by SD-OCT with A-FT ≤190 µm and B-FT >190 µm at the two moments (baseline = T0, and after 6 months = T180) considering that two eyes could be observed for one patient (patient as random effect). Also, the effect of the interaction between the group and time was evaluated.

A *p*-value <0.05 was considered statistically significant. All statistical analyses were done with STATA v14 (Collage Station, Texas, USA).

## 3. Results

A total of 21 eyes of 15 patients affected with RP, 9 males and 6 females (mean age 52.06 ± 19.31 years, range 19–86 years) were enrolled in the study (Table 1). The visual functional and anatomical parameters and the average values recorded at baseline (T0) and at 6 months (T180) after surgery are shown in Table 2.

Based on FT, 8 of the 21 eyes were classified in group A (FT ≤190 µm) and the remaining 13 were classified in group B (FT >190 µm). All 15 patients completed the 6-month follow-up and none had systemic complications intra-operatively and post-operatively throughout that period. Mean values of the intraocular pressure recorded before and after surgery did not change significantly. The mixed model results showed a significant difference between the two groups in close-up visus. Specifically, group A-FT showed mean higher values than the group with >190 µm (group effect *p* = 0.031). While group B-FT showed significantly higher mean values than group A-FT in central fovea thickness (Cµm), µm^3^, and average retinal thickness (Aµm^2^) (Table 2). In all models, the interaction Time/Group had no significant effect (Table 3).

The ophthalmologic evaluation included the measurement of visual acuity for far and near distance: BCVA measured by ETDRS charts at 4 m in logMAR units. Mean BCVA before the treatment was 1.02 ± 0.76 logMAR (20/200) in group A-FT (*n* = 8) and 0.47 ± 0.21 logMAR (20/200) in group B-FT (*n* = 13). Specifically, BCVA in group A-FT varied from 1.02 to 1.01 logMAR (+1.76%), and from 0.47 to 0.45 logMAR (+4.51%) in group B-FT (Figure 2).

No patient showed a reduction in BCVA at the 6-month follow-up. There was no statistically significant difference in visual acuity from baseline within the same group or between the two groups at 6 months (1.01 ± 0.77 vs. 0.45 ± 0.18, respectively). Percentage variation was lower in A (−1.76%) than in B (−4.43%).

Close-up visus in points (pts): At baseline, mean close-up visus was 25.88 ± 20.29 pts in group A-FT (8 eyes), and 15.15 ± 5.86 pts in group B-FT (13 eyes).

At the 6-month follow-up visit, it decreased to 26.13 pts in group A, whereas it increased to 12.00 pts in group B, showing that there was a trend towards significance in the latter group. Percentage variation was negative in A (−0.97%); conversely, it was greatly increased in B (+20.79%) (Figure 3).

The average threshold sensitivity by MY at baseline was 5.45 ± 6.79 dB in group A-FT (*n* = 8), and 3.15 dB ± 6.45 SD in group B-FT (*n* = 13). In the 6-month follow-up, it increased in both groups (6.29 dB ± 8.11 SD vs. 4.18 dB ± 7.79 SD, respectively).

Percentage improvement in retinal sensitivity was lower in group A (+15.41%) than in group B (+32.70). Despite the improvement in retinal sensitivity, it was not significant within the same group or between the two groups (Figure 4, Figure 5 and Figure 6).

Surveying the subjective experience of all patients at 6 months post-surgery with patient compliance analysis, it was reported that visual performances improved in 15 out of 21 eyes (71.43%), were unchanged in 4 eyes (19.05%), and worse in 2 eyes (9.52%) (Table 4).

However, examining patient feedback according to foveal thickness, the perception of improvement would be greater for patients with FT > 190 µm (11 eyes, 84.62%), rather than for patients with FT ≤ 190 µm (4 eyes, 50%) (Figure 7). If we considered the improved group alone, 11 eyes (73.33%) belonged to group B, and 4 (26.67%) to group A (Figure 7).

## 4. Discussion

The main objectives of our suprachoroidal autograft technique were to evaluate whether autologous stem cell transplantation may be useful for retinal restoration through the paracrine secretion of factors promoting vascular pedicle fat engraftment with the underlying tissue, and by enhancing pedicle fat original vascularization to ensure its volume and survival. Furthermore, the secondary aim was to evaluate prognostic factors to identify the time and tests needed to allow appropriate surgical intervention in those affected with RP.

LRRT cell therapy has been proven to have an impact on certain functional parameters after interaction with the residual cells. Close-up visus and retinal sensitivity improved in group B-FT, in which foveal thickness was greater, compared to group A-FT, with thinner FT and lower cellularity. Results of our study cast light on the therapeutic potential of stem cell implant activity that therefore could be crucial for retinal degeneration. Given these findings, the group with a foveal thickness greater than 190 microns was associated with a better prognosis, while in patients with thinner FT, the low cellular concentration might hinder the alleged beneficial interactions between stem cell implants and membrane receptors. Hence, central thickness is an important parameter to understand the complex processes underlying RP progression.

The myriad of bioactive factors released by the graft of three different types of autologous cells could be as follows:(1)Fat cells, which are contained in the pedicle grafted into the suprachoroidal space, secrete basic fibroblast GF (bFGF), interleukin (IL), epidermal GF (EGF), transforming GF (TGF), pigment epithelium-derived factor (PEDF), insulin-like GF-1 (IGF-1), and adiponectin [36,37,38].(2)ADSCs secrete bFGF, vascular endothelial GF (VEGF), granulocyte-macrophage colony-stimulating factor (GM-CSF), macrophage colony-stimulating factor (M-CSF), TGF, hepatocyte GF, IGF-1, IL, angiogenin, placental GF (PlGF), ciliary neurotrophic factor (CNTF), and brain-derived neurotrophic factor (BDNF) [39,40].

PLTs secrete platelet-derived GF (PDGF), VEGF, bFGF, TGF, EGF, IGF-1, platelet-derived angiogenesis factor (PDAF), and thrombospondin (TSP) [41,42]. Hence, the rationale behind this autograft lies in exploiting the stabilizing effect exerted by cytokines and GFs released by the grafted cells. Direct contact of the autograft with the choroid enhances the incretion of these bioactive actors into the choroidal flow, and consequently favors their dissemination throughout the retinal tissue and in the vitreous body.

GF binding to its own specific receptor in the target cell is the initial step that triggers an intracellular signaling transduction cascade, activating second messengers. The latter can activate specific intracellular biochemical pathways generally by a series of phosphorylation events, with the ultimate aim of regulating enzyme activity or gene expression [43,44].

Notably, the activated transcription factors, entering the nucleus and binding directly or indirectly to DNA, could regulate the expression of various genes with different mechanisms, promoting an increased synthesis of proteins, including enzymes and cytokines [32].

The significance of stem cell implants lies in their essential role of cell cycle regulation, since their presence could trigger the cell transition from G_0_ or quiescent phase to G_1_ or growth phase, which is necessary to enter the cellular growth cycle. Moreover, they are also important for stimulating a wide range of cellular processes, including mitosis, cell survival, migration, and cellular differentiation [45].

Mesenchymal cell graft into the sovrachoroidal space should promote a continuous incretion of GFs that are capable of interfering with the evolution of RP in several ways: Antioxidant, antinflammatory, antiapoptotic, citoprotective, and hemorheological activities [46,47].

Antioxidant activity. The bFGF and BDNF concentration within the photoreceptors has been shown to increase in response to stress in order to promote retinal cell survival and to prevent oxygen-induced photoreceptor cell death in the posterior retina. [48,49,50,51,52]. Moreover, rod survival is essential for extending the life span of cones inasmuch as the paracrine secretion of rod–cone viability factor (RdCVF) by rods is a pivotal trophic factor for cone survival [53,54]. It has been demonstrated that RdCVF has an antioxidant activity, and decreases cone death in rd10 and P23H transgenic rat models [55].

Antinflammatory activity. Several studies have reported that the activation of microglia generally occurs simultaneously or just before the peak of apoptotic photoreceptor death in RP [56,57]. The eye is an immune-privileged organ, and microglia and RPE cells are the front line of retinal immune defense [58]. Not only does RPE perform a number of processes essential for retinal homeostasis and function, but RPE cells are capable of secreting a diversified panel of proinflammatory cytokines, e.g., IL-6, IL-8, monocyte chemoattractant protein-1, and interferon-β (IFN-β), as well as anti-inflammatory factors, such as IL-11 and TGF-β [59,60,61]. Furthermore, microglial cells normally exist in a quiescent state until they are activated by the debris of dead or apoptic cells, lipopolysaccharides, or reactive oxygen species (ROS) during the course of RP [62,63]; they express a unique set of proinflammatory cytokines and chemokines [64,65]. In addition, intravitreal administration of MSC has been shown to have a remarkable effect on the host immune response by suppressing proinflammatory cytokine production, such as IFN-β and tumor necrosis factor-α through IL-1 receptor antagonist, and prostagandin E2 receptor activation [37]. Another study by Guadagni et al. has shown that a microenvironment supplemented with GFs can slow down the genetically determined photoreceptor death, concurrently reducing retinal inflammation, and thereby establishing framework conditions for the viability of the overall cell population [17].

Antiapoptotic activity. Excess generation of ROS causes damage to membrane lipoproteins and cellular DNA, thus leading to apoptosis and photoreceptor death [66,67,68,69]. The GFs excreted by grafted mesenchymal cells can facilitate *Bcl-2* gene expression in order to avoid the unrelenting cell death [21]. Bcl-2 family proteins are most notable for their regulation of apoptosis by interacting with caspases [70,71,72,73,74,75]. More specifically, the process is orchestrated by regulatory cytokines by either inhibiting or inducing apoptosis by blocking inhibitory mediators [75,76]. The latter process could be avoided, or at least delayed, by the anti-apoptotic activation of the *Bcl-2* gene induced by GFs derived from implanted mesenchymal cells. Basically, these factors replace those that should have been produced by retinal cells, which are quantitatively reduced and functionally impaired due to RP [66,68,71].

Citoprotective activity. In rat models with inherited retinal dystrophy, it has been shown that MSC contributes to visual function by the putative paracrine release of trophic cytokines that promote the clearance of dysmetabolic products of photoreceptors by RPE phagocytes [35]. Data from another similar study provide evidence that neurotrophic factors, i.e., bFGF, PEDF, nerve GF released by adipose tissue-derived MSCs, are involved in ensuring the survival of both retinal ganglion cells and photoreceptors [77,78]. In addition, VEGF released by PRP has been shown to stimulate the proliferation of ADSCs that hence promote the survival of grafted autologous fat and adipocytes [79].

Hemorheological activity. The progressive photoreceptor loss that occurs in RP has been identified as the cause of microvascular dysfunction due to the release of cellular waste products secondary to apoptosis. In this case, as well, the ensuing altered perfusion may end up in a vicious circle, leading to the final loss of photoreceptors [80]. Decreased choroidal blood flow is now known to induce dysfunction of visual sensitivity [81]. Research publications across different study settings support that blood flow is decreased in RP. By proper monitoring of intraocular pressure, Langham and Kramer highlighted the association between choroidal ischemia and visual loss, as well as RPE cell degeneration in RP patients [82]. Beutelspacher et al. found that retinal blood flow is lower in RP patients than the control group, thus concluding that the ensuing reduction of retinal vessels is a typical feature of RP [83]. Turksever et al. demonstrated that retinal oxygen uptake in RP patients is decreased, having found increased venous oxygen saturation in the case group [84]. Ayton et al. and Murakami et al. showed that RP patients had a thinner choroid than the control group, and observed that those patients were characterized by reduced visual acuity, thereby assuming that the choroidal thickness in RP can be a potential predictor of the therapeutic outcome [85,86]. Several factors, such as VEGF, bFGF, angiogenin, PDAF, PlGF, PDGF, EGF, and TGF-β, have been shown to promote endothelial regeneration and may therefore contribute to reperfusion of the choriocapillaris [87,88,89]. PLTs, primarily known for their contribution to hemostatis, are also able to release factors that promote tissue repair and regeneration and angiogenesis [5,41]. PRP acts as a trigger for the early development of a new capillary plexus, facilitating oxygen and nutrient diffusion towards the grafted cells [87,88,90].

Our research presents limits and critical points that will have to be addressed later to clarify some concepts and deepen the study on patients treated with the exposed technique. The following will be needed:(1)A greater number of patients and operated eyes with homogeneous age range;(2)Longitudinal studies to evaluate the longevity of the grafted tissue;(3)Biomolecular studies to understand the paracrine increment of the autograft;(4)Genetic tests obtained from patients to allow differentiation in homogeneous research groups, as genetic diagnosis will surely become more relevant in coming years, and it will be possible to determine the impact of MSC administration on different genetic groups of RP patients;(5)Evaluation of the suitable time in which the autograft must be performed in order to avoid failure in the presence of markedly degenerated retinal tissue. In particular, the results of this study show that FT might be considered a prognostic criterion for RP patients undergoing treatment by LRRT.

## 5. Conclusions

In light of the above, we can affirm that autologous transplants implanted in our RP patients at the ocular level constitute scientific evidence recognized by the aforementioned studies. However, we are aware that we are still in an experimental phase that will have to be deepened with numerous studies.

## Figures and Tables

**Figure 1 biomedicines-07-00094-f001:**
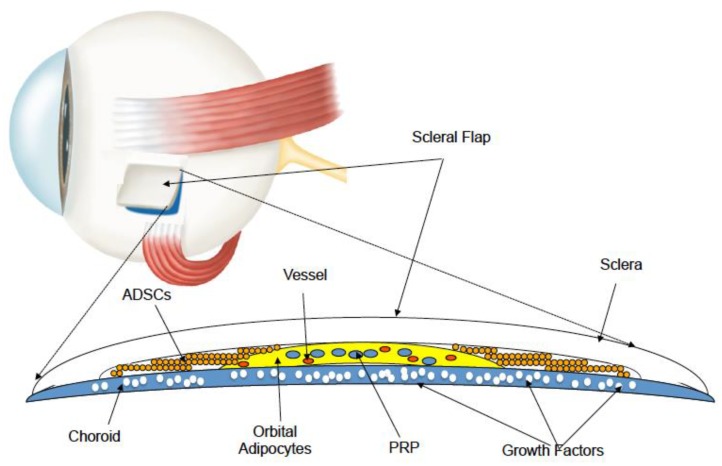
The suprachoroidal autograft obtained by the Limoli retinal restoration technique (LRRT) allows placing adipose stromal cells, adipose tissue-derived stem cells (ADSCs), and platelets (PLTs), obtained from PLT-rich plasma (PRP), close to the choroid. The production of growth factors (GFs), typical of these cells is poured directly into the choroidal flow, helping to maintain retinal cell trophism.

**Figure 2 biomedicines-07-00094-f002:**
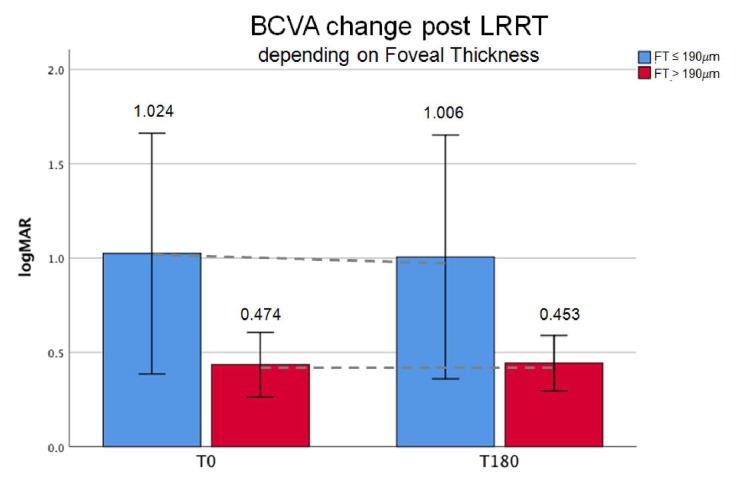
The best corrected visual acuity (BCVA), in logarithm of the minimum angle of resolution (logMAR) units, was stable after suprachoroidal autograft or increased (+4.51%) in patients with foveal thickness (FT) >190 µm (13 eyes) (B-FT group, green bars). LRRT: Limoli retinal restoration technique; T0: Baseline; T180: At 6 months from surgery. A-FT group with FT ≤190 µm (8 eyes, blue bars).

**Figure 3 biomedicines-07-00094-f003:**
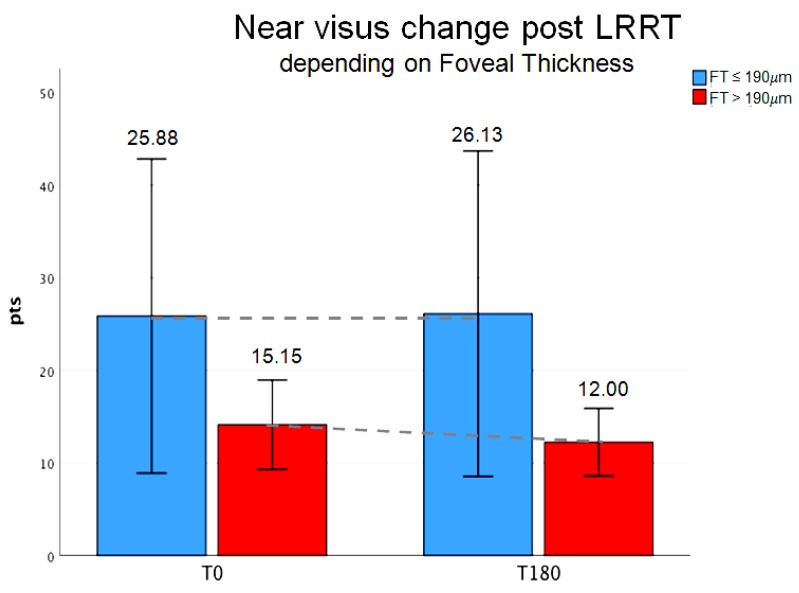
Close-up visus, in points (pts), change post-Limoli retinal restoration technique (LRRT) depending on foveal thickness (FT). Six months after surgery (T180) from the baseline (T0), close-up visus was stable in group A-FT (FT ≤ 190 µm, blue bars) and increased in group B-FT (FT > 190 µm, green bars). The increase was + 20.79%, corresponding to useful reading area (6–10 pts: Book, journal, etc.). Average at T0 was 25.88 (±20.28 SD) and at T180 was 26.13 (±21.03 SD) in group A-FT. Average at T0 was 15.15 (±5.85 SD) and at T180 was 12.00 (±4.00 SD) in group B-FT.

**Figure 4 biomedicines-07-00094-f004:**
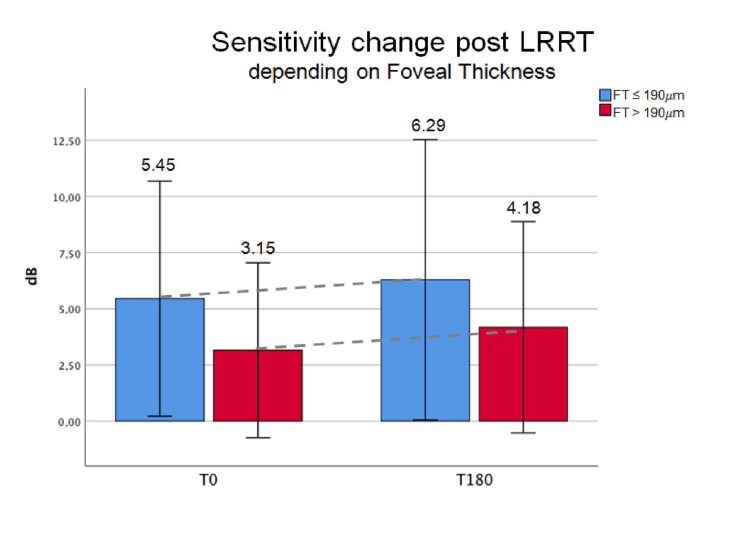
At 6 months (T180) from Limoli retinal restoration technique (LRRT), there was a more relevant to, +32.70%, for sensitivity in the group with foveal thickness (FT) > 190 µm (B-FT, green bars). Sensitivity in group A-FT (blue bars) was +15.41%. Average at T0 was 5.45 (±6.79 SD) and at T180 was 6.29 (±8.10 SD) in group A-FT. Average at T0 was 3.15 (±6.44 SD) and at T180 was 4.18 (±7.78 SD) in group B-FT.

**Figure 5 biomedicines-07-00094-f005:**
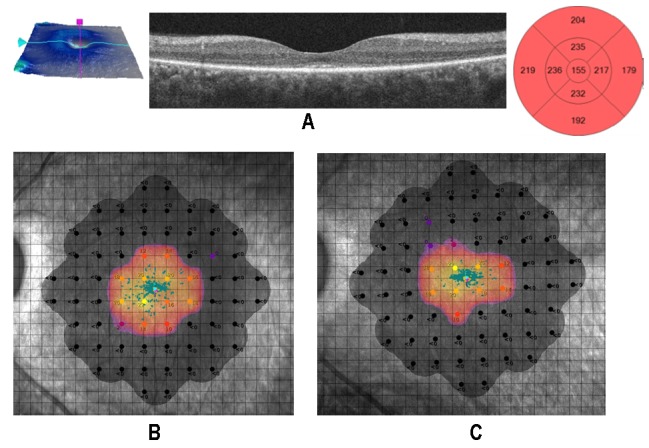
Retinitis pigmentosa (RP) patient with thinner foveal thickness (FT) < 190 µm (group A-FT). (**A**) The retinal cell population is small, foveal structures are often dystrophic, and the photoreceptor/retinal pigment epithelium/Bruch’s membrane/choriocapillaris complex is no longer recognizable. (**B**,**C**) The microperimetric sensitivity after surgery changed from 2 to 1.4 dB, and best corrected visual acuity (BCVA) changed from 0.097 to 0.155 logarithm of the minimum angle of resolution (logMAR).

**Figure 6 biomedicines-07-00094-f006:**
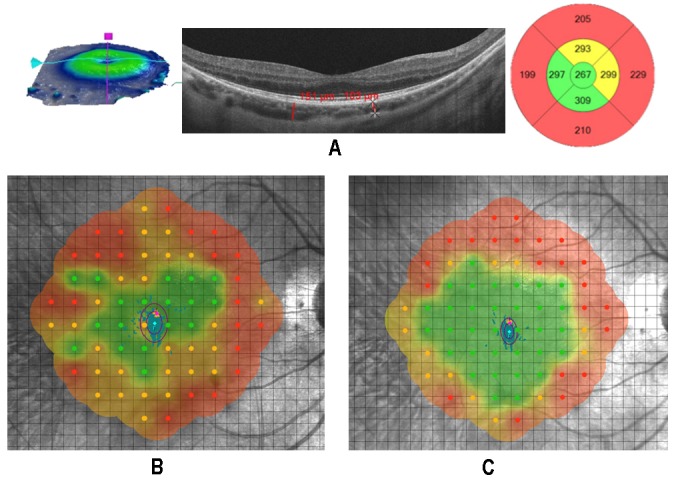
Retinitis pigmentosa (RP) patient with foveal thickness (FT) > 190 µm (group B-FT). (**A**) The retinal cell population is large, foveal structures are still intact, and the photoreceptor/retinal pigment epithelium/Bruch’s membrane/choriocapillaris complex is recognizable. (**B**,**C**) The microperimetric sensitivity after surgery changed from 14.41 to 16.61 dB and bivariate contour ellipse area (BCEA) (see central oval circles), used for fixation stability evaluation, changed from 2.0 to 0.9 using microperimetry (MY) device. Best corrected visual acuity (BCVA) changed from 0.045 to 0.000 logmar of the minimum angle of resolution (logMAR).

**Figure 7 biomedicines-07-00094-f007:**
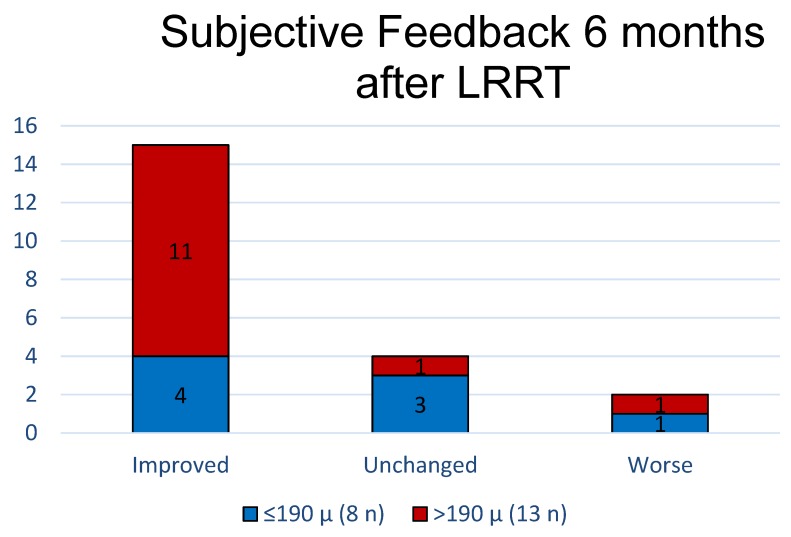
Retinitis pigmentosa (RP) patient compliance analysis at 6 months post-surgery depending on foveal thickness (FT). Compliance was good in 71.43% of all cases (groups A-FT and B-FT). Patients reported seeing better, but the percentage reached 84.62% in those with FT > 190 µm. In the improved group, 11 eyes (73.33%) belonged to group B-FT (green bars), and 4 (26.67%) to group A-FT (blue bars).

**Table 1 biomedicines-07-00094-t001:** Demographic data of retinitis pigmentosa (RP) patients with foveal thickness (FT) ≤ 190 µm (A-FT) and >190 µm (B-FT).

Patients	Group A-FT ≤ 190 µm	Group B-FT > 190 µm	Total
Number: Patients/eyes	6/8	9/13	15/21
Age years (± standard deviation (SD))	40.33 (13.98)	59.88 (18.93)	52.06 (19.31)
Range (years)	19–54	32–86	21–82
Female/male	3/3	3/6	6/9
Eye: Right/left	2/6	7/6	9/12

**Table 2 biomedicines-07-00094-t002:** Descriptive characteristics of analyzed parameters in the two groups according to the foveal thickness (FT): A-FT ≤190 µm (*n* = 8) and B-FT >190 µm (*n* = 13), at baseline (T0) and at 6 months (T180); mixed model results.

Parameters	Group	Mean ± SD Min–Max	Values (T0)	Values (T180)	%	Time Effect *p*-Value	Group Effect *p*-Value
logMAR	A-FT ≤190 µm	mean ± SD	1.02 ± 0.76	1.01 ± 0.77	+1.76		
		min–max	0.10–2.70	0.10–2.70			
	B-FT >190 µm	mean ± SD	0.47 ± 0.21	0.45 ± 0.18	+4.51		
		min–max	0.15–0.70	0.15–0.79		0.562	0.051
pts	A-FT ≤190 µm	mean ± SD	25.88 ± 20.29	26.13 ± 21.03	+0.97		
		min–max	8–64	7–64			
	B-FT >190 µm	mean ± SD	15.15 ± 5.86	12 ± 4	+20.79		
		min–max	7–26	7–18		0.269	0.08
dB MAIA	A-FT ≤190 µm	mean ± SD	5.45 ± 6.8	6.29 ± 8.11	+15.41		
		min–max	0–16	0–18.2			
	B-FT >190 µm	mean ± SD	3.15 ± 6.45	4.18 ± 7.79	+32.70		
		min–max	0–19.4	0–21.8		0.003	0.535
Cµm	A-FT ≤190 µm	mean ± SD	140.75 ± 37.42	133.88 ± 54.28	−0.05		
		min–max	49–160	0–161			
	B-FT >190 µm	mean ± SD	275.46 ± 88.1	275.08 ± 89	0.00		
		min–max	195–462	187–471		0.303	<0.001
µm^3^	A-FT ≤190 µm	mean ± SD	7.03 ± 1.39	7.67 ± 0.45	+0.09		
		min–max	4.6–8.7	7.3–8.6			
	B-FT >190 µm	mean ± SD	8.92 ± 1.38	8.79 ± 1.48	−0.01		
		min–max	6.5–10.7	6.5–11		0.806	0.023
Aµm^2^	A-FT ≤190 µm	mean ± SD	202.49 ± 23.4	212.86 ± 12.75	+0.05		
		min–max	164.9–240	202–239			
	B-FT >190 µm	mean ± SD	247.62 ± 38.69	244.15 ± 40.4	-0.01		
		min–max	179–299	181–305		0.949	0.023

LogMAR: Logarithm of the minimum angle of resolution; pts: Close-up visus in points; dB MAIA: Microperimetric sensitivity in deciBel. Cµm: Thickness of central fovea (in µm); µm^3^: Volume area; Aµm^2^: Average of retinal thickness (in µm); SD: Standard deviation.

**Table 3 biomedicines-07-00094-t003:** Variation between time at baseline (T0) and at 6 months (T180) estimated by mixed model in two groups according to the foveal thickness (FT): A-FT ≤190 µm (8 eyes) and B-FT >190 µm (13 eyes).

Variation (T180–T0)		A-FT ≤190 µm 8 Eyes	B-FT >190 µm 13 Eyes	Interaction Effect *p*-Value
logMAR	mean ± SD	−0.02 ± 0.07	−0.02 ± 0.04	0.971
pts	mean ± SD	0.25 ± 3.76	−3.15 ± 1.24	0.390
dB MAIA	mean ± SD	0.84 ± 0.59	1.02 ± 0.53	0.818
Cµm	mean ± SD	−6.88 ± 6.71	−0.38 ± 1.59	0.346
µm^3^	mean ± SD	0.35 ± 0.37	−0.12 ± 0.18	0.248
Aµm^2^	mean ± SD	5.66 ± 5.63	−3.46 ± 4.66	0.212

LogMAR: Logarithm of the minimum angle of resolution; pts: Close-up visus in points; dB MAIA: Microperimetric sensitivity in deciBel; Cµm: Thickness of central fovea (in µm); µm^3^: Volume area; Aµm^2^: Average of retinal thickness (in µm); SD: Standard deviation.

**Table 4 biomedicines-07-00094-t004:** Compliance analysis at 6 months (T180) post-surgery in two groups according to the foveal thickness (FT): A-FT ≤ 190 µm, and B-FT > 190 µm.

Compliance	A-FT ≤ 190 µm (8 Eyes)		B-FT > 190 µm (13 Eyes)	
Improved	4	50.00%	11	84.62%
Unchanged	3	37.50%	1	7.69%
Worse	1	12.50%	1	7.69%

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
