# Peer review of "Stem Cell Surgery and Growth Factors in Retinitis Pigmentosa Patients: Pilot Study after Literature Review"

_biomedicines, 2019, doi:10.3390/biomedicines7040094_

Round 1

Reviewer 1 Report

The authors of the article entitled "Stem cell surgery and growth factors in Retinitis Pigmentosa: Pilot study after literature review" has resubmited their work on implantation of autografts into the retina of RP patients to provide certain nutrients and growth factors.

In this new version authors have significantly increase the quality of the paper. For instance, the discussion it is not anymore an extensive review, although some parts still look like. However there are still some points of attention:

1) Paragraphing is still a problem. There are several paragraphs in the intro that can be combined. For instance line 16 and 17 of page 2. The aim is... This objective can be achieve (it is clearly related), so why very short paragraphs of even one single sentence? Please readjust because it makes the reading to some extent a bit annoying. Of note, this comment was indicated before and authors stated that they considered this request, but still, I think it is not enough.

2) i just noticed that none of the graphs have error bars. I can understand that the ones that the numbers involve patients cannot have them. However, those including measurements should definitely include the standard deviation or standard error. And this should be indicated in the figure legend. Also, Fig 3, Fig 4 and Fig 7 miss the color legend in the graph, in addition, Fig 4 misses the color explanation also in the figure legend.

3) Discussion is still a bit vague, as indicated before there are many things to be discussed. I am still missing more on the genetics, and whether this is useful to everyone. Patients reading this article may think that this can be a solution for every single RP patient. So we, as scientists, need to be careful with the way we say things to avoid misinterpretation.

4) English needs to be recheck. Beside the paragraphs, there are typo's every now and then.

5) Please indicate all abbreviations in the table of abbreviations, but also in the text when you first mention them. For instance SVF is not even at the abbreviation section, and others abbreviations are not in the text but since the abbreviations are at the end, reader try to look to the first pages to try to find the meaning of certain words.

Regarding the answers of the authors:

1) Please indicate the 15 and 21 eyes in a clear way in the text. Although being this a pilot I have my doubts whether it is ethical to inject the second eye just because the patient asks for it. Please provide the ethical approval for the second eye injection.

2) comment about the longevity of the treatment and other points of discussion. I appreciate that authors appreciate the comment, but it would be good to discuss in the paper this. Of course authors can hypothesise, but also can be critical and indicate the limitations and the points of attention for next studies.  

Author Response

Rome 21 October, 2019

Dear Editor-in-Chief
Biomedicines Journal,

We are sending You a review of the manuscript as recommended by Eastern Zhuang (Manuscript ID: Biomedicines- 602545 - Revision):
”Stem Cell Surgery and Growth Factors in Retinitis Pigmentosa Patients: Pilot Study After Literature Review”, by Paolo Giuseppe Limoli, Enzo Maria Vingolo, Celeste Limoli, and Marcella Nebbioso, that has been reviewed according to the Editor and Referees’ comments.

Changes made in response to comments are included at the bottom of this letter.

We are grateful to the Reviewers for their valuable advice in order to improve our paper.

Your sincerely

                                      Marcella Nebbioso & Co-Authors   

Address of Corresponding Author

Nebbioso Marcella, M.D. v.le del Policlinico, 155, 00161, Rome, Italy.

Phone: ++39/06/49975422    Fax: ++39/06/49975425.

marcella.nebbioso@uniroma1.it

Review Report Form Reviewer #1: 

Comments and Suggestions

1) Paragraphing is still a problem. There are several paragraphs in the intro that can be combined. For instance line 16 and 17 of page 2. The aim is... This objective can be achieve (it is clearly related), so why very short paragraphs of even one single sentence? Please readjust because it makes the reading to some extent a bit annoying. Of note, this comment was indicated before and authors stated that they considered this request, but still, I think it is not enough.

The entire manuscript has been revised and modified for a more fluent reading.

2) i just noticed that none of the graphs have error bars. I can understand that the ones that the numbers involve patients cannot have them. However, those including measurements should definitely include the standard deviation or standard error. And this should be indicated in the figure legend. Also, Fig 3, Fig 4 and Fig 7 miss the color legend in the graph, in addition, Fig 4 misses the color explanation also in the figure legend.

Error bars, figures and legends have been integrated.

3) Discussion is still a bit vague, as indicated before there are many things to be discussed. I am still missing more on the genetics, and whether this is useful to everyone. Patients reading this article may think that this can be a solution for every single RP patient. So we, as scientists, need to be careful with the way we say things to avoid misinterpretation.

This is a very important observation and we want to give only useful information to the research. Since RP contains at least 80 different genetic types would make it very difficult at the moment to have an adequate number of homogeneous cases even for a pilot study. Obviously, the next research will be performed taking into account the genetic data.

4) English needs to be recheck. Beside the paragraphs, there are typo's every now and then.

We have delivered the manuscript to a translator of mother tongue for corrections. We are waiting for the English language corrections and for the moment we send the manuscript. 

5) Please indicate all abbreviations in the table of abbreviations, but also in the text when you first mention them. For instance SVF is not even at the abbreviation section, and others abbreviations are not in the text but since the abbreviations are at the end, reader try to look to the first pages to try to find the meaning of certain words. 

The acronyms have been checked and corrected.

Regarding the answers of the authors:

1) Please indicate the 15 and 21 eyes in a clear way in the text. Although being this a pilot I have my doubts whether it is ethical to inject the second eye just because the patient asks for it. Please provide the ethical approval for the second eye injection.

Some patients have approved and signed the treatment in both eyes.

2) comment about the longevity of the treatment and other points of discussion. I appreciate that authors appreciate the comment, but it would be good to discuss in the paper this. Of course authors can hypothesise, but also can be critical and indicate the limitations and the points of attention for next studies.

We have also included the critical points and of attention in the Discussion Section. 

Comments and Suggestions for Authors: REVIEWER 2

This manuscript addresses an important problem and provides key information regarding the use of stem cells for retinal remodeling and repair. Although for the most part, the manuscript is well-written, the style of presentation format and use of bullet points should be revised to manuscript format.  The methods appear appropriate.  However, the manuscript suffers from small sample size, and wide age ranges (19-86 years), both of which contribute to the lack of significance in outcomes.  Considering that this is a pilot study with potential, a larger clinical trial with an appropriate samples size, and stratification by age, as well as biomolecular outcomes should be conducted.  The following are recommended:

The various critical points have been exposed in the manuscript.

Abstract:
a. Line 15: The authors state that the purpose was to evaluate whether the technique “can produce growth factors”….  Yet no data are provided regarding growth factors to demonstrate that the objective was achieved.  Thus, this sentence should be restated as presented in lines 7-9 on page 3.  
b. The conclusions should be also be restated.  As it is written, it gives the impression that the conclusions are based on in-depth review of other studies and not the current study by the authors.  The conclusions should provide a take-home message that addresses the objectives of the study.

In the Abstract Section the sentence has been reformulated.

The Conclusions now provide a message to take home in relation to the objectives of the study (see page 14).

Introduction:  

Page 2, line 10: The authors state “new therapeutic options are being activin developed”.... provide references at the end of the sentence. [9-12] Same comment for lines 30-32, “have generated much controversy”.

We have inserted the references.

Page 3, line 7: this appears to be a “prospective”, not “retrospective” study since the patients were recruited prior to the surgical procedures.

The sentence "prospective study" has been inserted.

Page 3, lines 9-11: Clarify whether this is the working hypothesis.  

The objective of the research has been clarified (see page 3).

Materials & Methods: Page 3, lines 21-22: the sentence should be revised to clearly state that the project was approved by the Ethics committee, not just the informed consent.  Signed written informed consent was obtained by the patients, but the protocol itself should be approved by the Ethics committee.  Also, provide the approval number of the protocol.

We have included the information requested at page 3.

Page 4, line 21: change “ FT thicker” to “thicker FT”. Page 4, line 23: clarify “panel A”.  Page 4, line 32: add “the” between “in” and “literature”. Page 4, line 36: change “5’” to 5 minutes.

Results:
Page 7, line 24 states that “there was a trend towards significance in group B for close-up visus”.  However, due to this lack of significance, the authors should revise the abstract on line 19 to clearly state “non-significant improvement”.

Discussion:
a. The discussion is written in a presentation style.  It should be re-written in manuscript style without bullets.

Page 11, lines 8-13.  These objectives appear different from those stated in the introduction.

We have corrected the errors.

Throughout the discussion, the authors use growth factors, when no growth factors or biomolecular assessments were made.  In order to prevent confusion to the reader, the authors should correctly state “stem cell implants” or a similar terminology, as appropriate.

We replaced GF with the sentence “stem cell implants”.

No substantive explanation was provided for the differences in outcomes between the two groups. The discussion should be re-written to focus on the data and outcomes of the study, and not as a review of the literature.

We have considered the central retinal thickness as a cause of poor significance of the result.  Moreover the inhomogeneous group of work and the lack of molecular and genetic data allow us to criticize our study. So we deepened the literature in the hope of continuing to work by improving the results with a short-term search.

The conclusions do not adequately address the objectives stated on page 3.

We have modified the Conclusions in this sense.

Reviewer 2 Report

Brief Overview:

Limoli PG et al. evaluated the efficacy the Limoli retinal restoration technique (LRRT) which utilizes an implant of cell types of mesenchymal origin such as adipose stromal cells, adipose tissue-derived stem cells, and platelets for retinal restoration in 21 patients with retinitis pigmentosa (RP). Data were stratified between patients with <190 µm (group A) and >190 µm (group B) retinal foveal thickness. At 6 months follow-up, group B be showed nonsignificant improvements in close-up vision and sensitivity.  The authors conclude that transplant of mesenchymal stem cells could sustain neuroenhancement in patients with foveal thickness.

Overall Comments:

This manuscript addresses an important problem and provides key information regarding the use of stem cells for retinal remodeling and repair. Although for the most part, the manuscript is well-written, the style of presentation format and use of bullet points should be revised to manuscript format.  The methods appear appropriate.  However, the manuscript suffers from small sample size, and wide age ranges (19-86 years), both of which contribute to the lack of significance in outcomes.  Considering that this is a pilot study with potential, a larger clinical trial with an appropriate samples size, and stratification by age, as well as biomolecular outcomes should be conducted.  The following are recommended:

Abstract:
a. Line 15: The authors state that the purpose was to evaluate whether the technique “can produce growth factors”….  Yet no data are provided regarding growth factors to demonstrate that the objective was achieved.  Thus, this sentence should be restated as presented in lines 7-9 on page 3. 
b. The conclusions should be also be restated.  As it is written, it gives the impression that the conclusions are based on in-depth review of other studies and not the current study by the authors.  The conclusions should provide a take-home message that addresses the objectives of the study.

Introduction: 
a. Page 2, line 10: The authors state “new therapeutic options are being activin developed”.... provide references at the end of the sentence.

b. Same comment for lines 30-32, “have generated much controversy”.

c. Page 3, line 7: this appears to be a “prospective”, not “retrospective” study since the patients were recruited prior to the surgical procedures.

d. Page 3, lines 9-11: Clarify whether this is the working hypothesis. 

Materials & Methods:
a.       Page 3, lines 21-22: the sentence should be revised to clearly state that the project was approved by the Ethics committee, not just the informed consent.  Signed written informed consent was obtained by the patients, but the protocol itself should be approved by the Ethics committee.  Also, provide the approval number of the protocol.

b.  Page 4, line 21: change “ FT thicker” to “thicker FT”.

c.       Page 4, line 23: clarify “panel A”. 

d.      Page 4, line 32: add “the” between “in” and “literature”.

e.       Page 4, line 36: change “5’” to 5 minutes.

Results:
Page 7, line 24 states that “there was a trend towards significance in group B for close-up visus”.  However, due to this lack of significance, the authors should revise the abstract on line 19 to clearly state “non-significant improvement”.

Discussion:
a. The discussion is written in a presentation style.  It should be re-written in manuscript style without bullets.

b. Page 11, lines 8-13.  These objectives appear different from those stated in the introduction.

c. Throughout the discussion, the authors use growth factors, when no growth factors or biomolecular assessments were made.  In order to prevent confusion to the reader, the authors should correctly state “stem cell implants” or a similar terminology, as appropriate.

d. No substantive explanation was provided for the differences in outcomes between the two groups.

e. The discussion should be re-written to focus on the data and outcomes of the study, and not as a review of the literature.

f. The conclusions do not adequately address the objectives stated on page 3. 

Author Response

Rome 21 October, 2019

Dear Editor-in-Chief
Biomedicines Journal,

We are sending You a review of the manuscript as recommended by Eastern Zhuang (Manuscript ID: Biomedicines- 602545 - Revision):
Stem Cell Surgery and Growth Factors in Retinitis Pigmentosa Patients: Pilot Study After Literature Review”, by Paolo Giuseppe Limoli, Enzo Maria Vingolo, Celeste Limoli, and Marcella Nebbioso, that has been reviewed according to the Editor and Referees’ comments.

Changes made in response to comments are included at the bottom of this letter.

We are grateful to the Reviewers for their valuable advice in order to improve our paper.

Your sincerely

                                                  Marcella Nebbioso & Co-Authors   

Address of Corresponding Author

Nebbioso Marcella, M.D. v.le del Policlinico, 155, 00161, Rome, Italy.

Phone: ++39/06/49975422    Fax: ++39/06/49975425.

marcella.nebbioso@uniroma1.it

Review Report Form Reviewer #1:

Comments and Suggestions

1) Paragraphing is still a problem. There are several paragraphs in the intro that can be combined. For instance line 16 and 17 of page 2. The aim is... This objective can be achieve (it is clearly related), so why very short paragraphs of even one single sentence? Please readjust because it makes the reading to some extent a bit annoying. Of note, this comment was indicated before and authors stated that they considered this request, but still, I think it is not enough.

The entire manuscript has been revised and modified for a more fluent reading.

2) i just noticed that none of the graphs have error bars. I can understand that the ones that the numbers involve patients cannot have them. However, those including measurements should definitely include the standard deviation or standard error. And this should be indicated in the figure legend. Also, Fig 3, Fig 4 and Fig 7 miss the color legend in the graph, in addition, Fig 4 misses the color explanation also in the figure legend.

Error bars, figures and legends have been integrated.

3) Discussion is still a bit vague, as indicated before there are many things to be discussed. I am still missing more on the genetics, and whether this is useful to everyone. Patients reading this article may think that this can be a solution for every single RP patient. So we, as scientists, need to be careful with the way we say things to avoid misinterpretation.

This is a very important observation and we want to give only useful information to the research. Since RP contains at least 80 different genetic types would make it very difficult at the moment to have an adequate number of homogeneous cases even for a pilot study. Obviously, the next research will be performed taking into account the genetic data.

4) English needs to be recheck. Beside the paragraphs, there are typo's every now and then.

We have delivered the manuscript to a translator of mother tongue for corrections. We are waiting for the English language corrections and for the moment we send the manuscript.

5) Please indicate all abbreviations in the table of abbreviations, but also in the text when you first mention them. For instance SVF is not even at the abbreviation section, and others abbreviations are not in the text but since the abbreviations are at the end, reader try to look to the first pages to try to find the meaning of certain words. 

The acronyms have been checked and corrected.

Regarding the answers of the authors:

1) Please indicate the 15 and 21 eyes in a clear way in the text. Although being this a pilot I have my doubts whether it is ethical to inject the second eye just because the patient asks for it. Please provide the ethical approval for the second eye injection.

Some patients have approved and signed the treatment in both eyes.

2) comment about the longevity of the treatment and other points of discussion. I appreciate that authors appreciate the comment, but it would be good to discuss in the paper this. Of course authors can hypothesise, but also can be critical and indicate the limitations and the points of attention for next studies.

We have also included the critical points and of attention in the Discussion Section.

Comments and Suggestions for Authors: REVIEWER 2

This manuscript addresses an important problem and provides key information regarding the use of stem cells for retinal remodeling and repair. Although for the most part, the manuscript is well-written, the style of presentation format and use of bullet points should be revised to manuscript format.  The methods appear appropriate.  However, the manuscript suffers from small sample size, and wide age ranges (19-86 years), both of which contribute to the lack of significance in outcomes.  Considering that this is a pilot study with potential, a larger clinical trial with an appropriate samples size, and stratification by age, as well as biomolecular outcomes should be conducted.  The following are recommended:

The various critical points have been exposed in the manuscript.

Abstract:
a. Line 15: The authors state that the purpose was to evaluate whether the technique “can produce growth factors”….  Yet no data are provided regarding growth factors to demonstrate that the objective was achieved.  Thus, this sentence should be restated as presented in lines 7-9 on page 3.  
b. The conclusions should be also be restated.  As it is written, it gives the impression that the conclusions are based on in-depth review of other studies and not the current study by the authors.  The conclusions should provide a take-home message that addresses the objectives of the study.

In the Abstract Section the sentence has been reformulated.

The Conclusions now provide a message to take home in relation to the objectives of the study (see page 14).

Introduction:  

Page 2, line 10: The authors state “new therapeutic options are being activin developed”.... provide references at the end of the sentence. [9-12] Same comment for lines 30-32, “have generated much controversy”.

We have inserted the references.

Page 3, line 7: this appears to be a “prospective”, not “retrospective” study since the patients were recruited prior to the surgical procedures.

The sentence "prospective study" has been inserted.

Page 3, lines 9-11: Clarify whether this is the working hypothesis.  

The objective of the research has been clarified (see page 3).

Materials & Methods:

Page 3, lines 21-22: the sentence should be revised to clearly state that the project was approved by the Ethics committee, not just the informed consent.  Signed written informed consent was obtained by the patients, but the protocol itself should be approved by the Ethics committee.  Also, provide the approval number of the protocol.

We have included the information requested at page 3.

Page 4, line 21: change “ FT thicker” to “thicker FT”. Page 4, line 23: clarify “panel A”.  Page 4, line 32: add “the” between “in” and “literature”. Page 4, line 36: change “5’” to 5 minutes.

Results:
Page 7, line 24 states that “there was a trend towards significance in group B for close-up visus”.  However, due to this lack of significance, the authors should revise the abstract on line 19 to clearly state “non-significant improvement”.

Discussion:
a. The discussion is written in a presentation style.  It should be re-written in manuscript style without bullets.

Page 11, lines 8-13.  These objectives appear different from those stated in the introduction.

We have corrected the errors.

Throughout the discussion, the authors use growth factors, when no growth factors or biomolecular assessments were made.  In order to prevent confusion to the reader, the authors should correctly state “stem cell implants” or a similar terminology, as appropriate.

We replaced GF with the sentence “stem cell implants”.

No substantive explanation was provided for the differences in outcomes between the two groups. The discussion should be re-written to focus on the data and outcomes of the study, and not as a review of the literature.

We have considered the central retinal thickness as a cause of poor significance of the result.  Moreover the inhomogeneous group of work and the lack of molecular and genetic data allow us to criticize our study. So we deepened the literature in the hope of continuing to work by improving the results with a short-term search.

The conclusions do not adequately address the objectives stated on page 3.

We have modified the Conclusions in this sense.

Round 2

Reviewer 1 Report

The paper by Limoli and colleagues describes a technique to deliver grafts to patients suffering from RP.

This is the second revision of the article and the quality of the reading, as well as figures, have improved. The authors have answered some of the questions and inserted most of the suggested changes.

I would like to indicate a couple of things that need to be improved:

1) the quality of the images is still poor, especially the graphs (at least in my version is blurry). 

2) Be consistent with the graphs, they all look different.

Last but not least, I would like to give advice to the authors:

The work of the reviewer is to review a paper. We do this for free, most of the time in our spare time and because we love science. So, if you need to ask for an extension, do not be afraid and ask for it, rather than submit another draft, while the final version is being prepared. Because the time of the reviewer is also precious, and if I have asked for an English revision, I am not going to revise something to write exactly the same comment again and have to review the manuscript again 2 weeks later. So please, facilitate also the work of the reviewer. I decided to contact the editor and ask for the final version corrected by a native person. Another person would have rejected the article. The comment above is related to this particular point:

4) English needs to be recheck. Besides the paragraphs, there are typo's every now and then.

We have delivered the manuscript to a translator of mother tongue for corrections. We are waiting for the English language corrections and for the moment we send the manuscript.

Author Response

Rome 14 November, 2019

Dear Editor-in-Chief

Biomedicines Journal,

We are sending You a review of the manuscript as recommended by Eastern Zhuang (Manuscript ID: Biomedicines- 602545 - Revision):

Stem Cell Surgery and Growth Factors in Retinitis Pigmentosa Patients: Pilot Study After Literature Review”, by Paolo Giuseppe Limoli, Enzo Maria Vingolo, Celeste Limoli, and Marcella Nebbioso, that has been reviewed according to the Editor and Referees’ comments.

Changes made in response to comments are included at the bottom of this letter.

We are grateful to the Reviewers for their valuable advice in order to improve our paper.

Your sincerely

                                                       Marcella Nebbioso & Co-Authors   

Address of Corresponding Author

Nebbioso Marcella, M.D. v.le del Policlinico, 155, 00161, Rome, Italy.

Phone: ++39/06/49975422    Fax: ++39/06/49975425.

marcella.nebbioso@uniroma1.it

Review Report Form Reviewer:

Comments and Suggestions

I would like to indicate a couple of things that need to be improved:

1) the quality of the images is still poor, especially the graphs (at least in my version is blurry). 

2) Be consistent with the graphs, they all look different.

Last but not least, I would like to give advice to the authors:

The work of the reviewer is to review a paper. We do this for free, most of the time in our spare time and because we love science. So, if you need to ask for an extension, do not be afraid and ask for it, rather than submit another draft, while the final version is being prepared. Because the time of the reviewer is also precious, and if I have asked for an English revision, I am not going to revise something to write exactly the same comment again and have to review the manuscript again 2 weeks later. So please, facilitate also the work of the reviewer. I decided to contact the editor and ask for the final version corrected by a native person. Another person would have rejected the article. The comment above is related to this particular point:

4) English needs to be recheck. Besides the paragraphs, there are typo's every now and then.

The Text and Figures have been revised and modified for a more fluent reading. English has been double checked.

Reviewer 2 Report

The authors have appropriately responded to all queries and concerns and the manuscript has been improved.  However, there are too many paragraphs in both the introduction and conclusion many of which can be combined.  In general, the introduction should be three paragraphs describing: 1) what is the problem; 2) what has been previously done about the problem, and what are the gaps in knowledge; and 3) what the current research aims to do to fill the gaps.  In this last paragraph the hypothesis and objectives are stated. 

In the conclusions, there are still bullets that should be deleted.   

Author Response

Rome 3 November, 2019
Dear Editor-in-Chief
Biomedicines Journal,

We are sending You a review of the manuscript as recommended by Eastern Zhuang (Manuscript ID: Biomedicines- 602545 - Revision):
”Stem Cell Surgery and Growth Factors in Retinitis Pigmentosa Patients: Pilot Study After Literature Review”, by Paolo Giuseppe Limoli, Enzo Maria Vingolo, Celeste Limoli, and Marcella Nebbioso, that has been reviewed according to the Editor and Referees’ comments.

Changes made in response to comments are included at the bottom of this letter.

We are grateful to the Reviewers for their valuable advice in order to improve our paper.

Your sincerely

                                                                                                Marcella Nebbioso & Co-Authors   

Address of Corresponding Author

Nebbioso Marcella, M.D. v.le del Policlinico, 155, 00161, Rome, Italy.

Phone: ++39/06/49975422    Fax: ++39/06/49975425.

marcella.nebbioso@uniroma1.it

Review Report Form Reviewer:

Comments and Suggestions

The authors have appropriately responded to all queries and concerns and the manuscript has been improved.  However, there are too many paragraphs in both the introduction and conclusion many of which can be combined.  In general, the introduction should be three paragraphs describing: 1) what is the problem; 2) what has been previously done about the problem, and what are the gaps in knowledge; and 3) what the current research aims to do to fill the gaps.  In this last paragraph the hypothesis and objectives are stated. 

The Introction and Conclusion sections have been revised and modified for a more fluent reading.